# De novo centriole formation in human cells is error-prone and does not require SAS-6 self-assembly

Won-Jing Wang[1]*, Devrim Acehan[2], Chien-Han Kao[1], Wann-Neng Jane[3], Kunihiro Uryu[2], Meng-Fu Bryan Tsou[4]*

[1]Institute of Biochemistry and Molecular Biology, College of Life Sciences, National Yang-Ming University, Taipei, Taiwan; [2]Electron Microscopy Resource Center, The Rockefeller University, New York, United States; [3]Institute of Plant and Microbial Biology, Academia Sinica, Taipei, Taiwan; [4]Cell Biology Program, Memorial Sloan-Kettering Cancer Center, New York, United States

**Abstract** Vertebrate centrioles normally propagate through duplication, but in the absence of preexisting centrioles, de novo synthesis can occur. Consistently, centriole formation is thought to strictly rely on self-assembly, involving self-oligomerization of the centriolar protein SAS-6. Here, through reconstitution of de novo synthesis in human cells, we surprisingly found that normal looking centrioles capable of duplication and ciliation can arise in the absence of SAS-6 self-oligomerization. Moreover, whereas canonically duplicated centrioles always form correctly, de novo centrioles are prone to structural errors, even in the presence of SAS-6 self-oligomerization. These results indicate that centriole biogenesis does not strictly depend on SAS-6 self-assembly, and may require preexisting centrioles to ensure structural accuracy, fundamentally deviating from the current paradigm.

*For correspondence: wangwj@ ym.edu.tw (WJW); tsoum@mskcc. org (MFBT)

**Competing interests:** The authors declare that no competing interests exist.

## Introduction

Centrioles are microtubule-based, ninefold symmetrical structures essential for centrosome and cilia formation. In cycling cells, centrioles are maintained in fixed numbers, and formed through canonical duplication depending on pre-existing (or mother) centrioles. In the absence of pre-existing centrioles, however, de novo synthesis can occur (*Khodjakov et al., 2002*). The number of centrioles formed through the de novo pathway is highly variable (*Khodjakov et al., 2002*; *La Terra et al., 2005*), providing an explanation for why canonical duplication dominates in dividing cells. In contrast to cycling cells, in post-mitotic cells such as multi-ciliated epithelia, the genes required for centriole assembly are highly up-regulated (*Hoh et al., 2012*) to produce large, variable numbers of centrioles prior to ciliogenesis, a process thought to primarily depend on de novo assembly (*Dirksen, 1991*). Interestingly, a recent study showed that the production of high quantities of centrioles in mouse multi-ciliated epithelia is in fact driven by the pre-existing centriole rather than through de novo assembly (*Al Jord et al., 2014*), suggesting that the presence of pre-existing centrioles may have additional roles other than the number control for centriole biogenesis.

Centriole biogenesis, canonical or de novo, starts with cartwheel assembly, a geometric scaffold that defines the shape and structural integrity of centrioles (*Anderson and Brenner, 1971*). The backbone of the cartwheel is characterized by a central hub from which nine spokes emanate (*Anderson and Brenner, 1971*) and is primarily made of the centriolar protein SAS-6 (*Kitagawa et al., 2011*; *van Breugel et al., 2011*). SAS-6 exists as dimers, which can self-oligomerize in vitro via an N-terminal head domain, forming a ring resembling the central hub, and C-terminal

**eLife digest** Cells pass on their characteristics or "traits" to new generations in the form of DNA molecules. DNA provides the instructions to make proteins, which may then assemble into larger structures without using any external templates in a process called self-assembly. However, when a cell divides, DNA is not the only element that is passed on to the daughter cells; many large protein structures that have assembled in mother cells are also divided between the daughter cells. The daughter cells may then produce extra copies of these protein structures, but it is not known whether the pre-existing structures are involved in this process.

Centrioles are complex structures made of proteins and play a crucial role in cell division. One of the main components of centrioles is a protein called SAS-6. Recent studies have shown that SAS-6 molecules can bind to each other to form "oligomers". This process, which is called self-oligomerization, has been proposed to drive the formation of centrioles.

Now, Wang et al. examine whether centrioles can form properly in cells when no other centrioles are present. The experiments show that centrioles can indeed form, but they are prone to structural errors. In contrast, centrioles that form in the presence of older centrioles are essentially free of errors. The experiments used human eye cells that were missing the gene that encodes SAS-6. These cells could not make centrioles, but when SAS-6 was re-introduced into these cells, new centrioles formed. Unexpectedly, re-introducing a mutant form of SAS-6 that cannot form oligomers into the cells still allowed new centrioles to form, which shows that self-oligomerization of SAS-6 is not essential for the assembly of centrioles.

Together, Wang et al.'s findings challenge the idea that SAS-6 self-oligomerization is involved in the formation of centrioles, and suggest that preexisting centrioles may help to minimize errors in the formation of new centrioles.

tails pointing outwards as spokes (*Kitagawa et al., 2011*; *van Breugel et al., 2011*; *van Breugel et al., 2014*), albeit not always ninefold symmetric in vitro(*Cottee et al., 2011*). Nevertheless, these elegant discoveries raise an exciting proposal that the self-assembly property associated with the N terminus of SAS-6 drives cartwheel and centriole formation.

In contrast to the SAS-6 self-assembly model, a template-based assembly model, dependent on the interaction of the C-terminal tail of SAS-6 with the lumen of mother centrioles, has recently been proposed to initiate canonical duplication (*Fong et al., 2014*). During S phase, SAS-6 molecules are first recruited to the proximal lumen of the mother centriole prior to centriole duplication, adopting a cartwheel-like organization through interactions with the luminal wall, rather than via their self-olig-omerization activity. This leads to a proposal that mother centrioles may function as the template to shape SAS-6 assembly, thereby preserving the geometric shape of the centriole that otherwise cannot be ensured by SAS-6 self-assembly alone. Notably, the template-based model appears incompatible with de novo centriole synthesis in which no pre-existing centrioles are required. However, as the nature of de novo synthesis, for example, whether it is indeed based on SAS-6 self-assembly, has not been determined, it is premature to accept or reject any of these ideas.

## Results

### Reconstitution of de novo centrosome synthesis in human cells

To characterize de novo formation of centrioles or centrosomes, we used CRISPR/Cas9 gene targeting to sequentially inactivate p53 and SAS-6 genes in retinal pigment epithelial cells (RPE1), generating stable, acentriolar cell lines ($SAS\text{-}6^{-/-}$; $p53^{-/-}$) in which de novo centrosome formation can be subsequently reconstituted (*Izquierdo et al., 2014*) (see 'Materials and methods'). Pure acentriolar cell lines were established through clonal propagation from single cells, a process taking 4–5 weeks, before these cells were used for experiments. In two independent $SAS\text{-}6^{-/-}$; $p53^{-/-}$ cell lines we generated (clone #1 and #2), frameshift mutations present in the beginning of the coding region were found in each SAS-6 allele (*Figure 1—figure supplement 1A*), predicting to produce severely truncated products consisting of 19 or 23 amino acids in clone #1 or #2, respectively. Consistently,

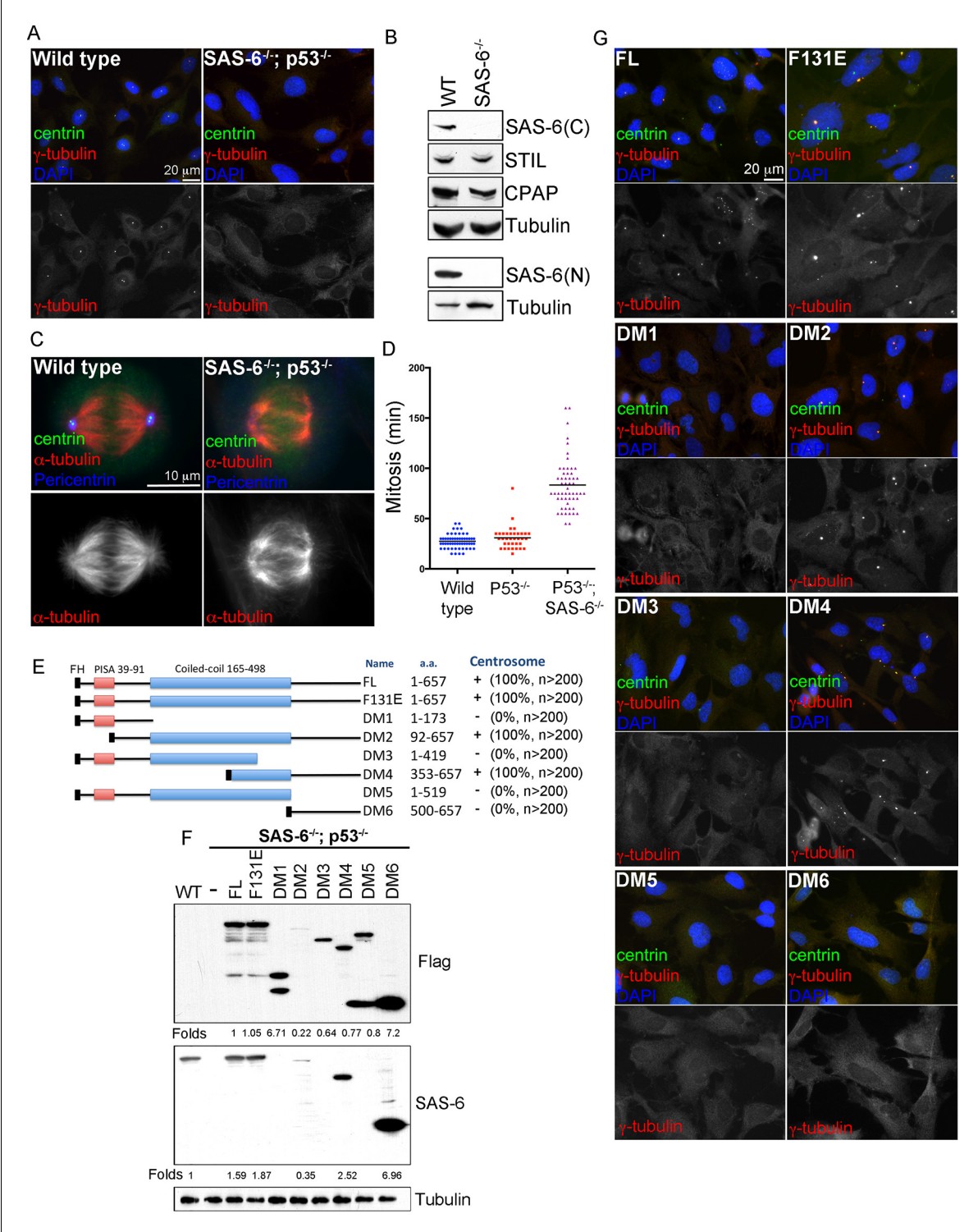

**Figure 1.** De novo centrosome formation in the absence of SAS-6 self-oligomerization. (**A**) Wild-type (WT) or acentrosomal (*p53*[-/-]; *SAS-6* [-/-]; clone #1) RPE1 cells were stained with anti-centrin (green) and γ-tubulin (red). DNA (DAPI, blue). (**B**) Western blot analysis of WT or SAS-6[-/-] cells with indicated antibodies, including both N- and C-terminal SAS-6 antibodies (SAS-6N; SAS-6C). (**C**) WT or acentrosomal cells in mitosis were stained with anti-centrin (green), α-tubulin (red), and pericentrin (blue). (**D**) The duration of mitosis in WT or SAS-6[-/-] cells was measured through time-lapse imaging of live cells. (**E**) A schematic diagram showing various SAS-6 mutants tagged with Flag and HA (FH). Clone #1 *p53*[-/-] *SAS-6*[-/-] cells were infected with lentiviruses carrying various of SAS-6 constructs, and the ability of each construct in rescuing centrosome formation was indicated and quantified in infected cells expressing detectable HA-tagged SAS-6. n, number of infected cells examined. (**F**) Isogenic *SAS-6*[-/-]; *p53*[-/-] cells carrying indicated SAS-6 constructs

*Figure 1. continued on next page*

*Figure 1. Continued*

(see Methods) were induced for SAS-6 expression for 3 days and then analyzed by western blot with Flag antibodies or C-terminal SAS-6 antibodies (SAS-6C). For analyses with N-terminal SAS-6 antibodies (SAS-6N), please see *Figure 1—figure supplement 1D*. Note that in Flag staining, DM6 leaked to the lane of DM5. The expression level of each SAS-6 construct, indicated as fold changes (Folds) relative to the endogenous SAS-6 or SAS-6$^{FL}$, is shown. (G) Isogenic *SAS-6$^{-/-}$*; *p53$^{-/-}$* cells carrying indicated SAS-6 constructs were treated as (F) and analyzed by immunofluorescence microscopy using indicated antibodies (Scale bar: 20 µm in A and G, 10 µm in C).

The following figure supplements are available for Figure 1:

**Figure supplement 1.** Characterization of *SAS6$^{-/-}$*; *p53$^{-/-}$* cell lines (clone #1 and #2).

**Figure supplement 2.** Induction of de novo centrosome formation in clone #2 *SAS6$^{-/-}$*; *p53$^{-/-}$* cells in the absence of SAS-6 self-oligomerization.

Western blot analyses using antibodies against either the N- or C-terminal region of SAS-6 failed to detect any SAS-6 signal in these cells (*Figure 1B*; *Figure 1—figure supplement 1C*). Similar frame-shift mutations were also seen in TP53 alleles (*Figure 1—figure supplement 1B*), leading to loss of p53 function (*Izquierdo et al., 2014*). Importantly, both SAS-6 knockout cell lines completely lack centrioles or centrosomes as expected (*Figure 1A* for clone #1; *Figure 1—figure supplement 2A* for clone #2), but can continue to proliferate in the absence of p53 (*Figure 1C* for clone #1; *Figure 1—figure supplement 2B* for clone #2) (*Bazzi and Anderson, 2014*; *Izquierdo et al., 2014*; *Lambrus et al., 2015*; *Wong et al., 2015*), although their M phase is significantly lengthened (*Figure 1D*). Intriguingly, when exogenous wild-type, full length SAS-6 (SAS-6$^{FL}$) was inducibly expressed in SAS-6$^{-/-}$ cells (see 'Materials and methods' for details), either in clone #1 or #2, variable numbers of centrosomes formed robustly in the absence of pre-existing centrosomes (*Figure 1E,G* for clone #1; *Figure 1—figure supplement 2C,D* for clone #2), a result consistent with previous reports (*Lambrus et al., 2015*; *Wong et al., 2015*). As clone #1 and clone #2 cell lines behave similarly, we used clone #1 to establish a stable, cell-based system in which the role of SAS-6 in de novo centrosome synthesis can be analyzed (see below).

## SAS-6 self-oligomerization is not required for de novo centrosome formation

To determine which domains of SAS-6 is required and sufficient for de novo centrosome formation, full length SAS-6 (FL) or various SAS-6 deletion mutants (DMs) were made to allow controlled expression under the doxycycline inducible promoter (*Figure 1E*). Isogenic, acentriolar *SAS-6$^{-/-}$*; *p53$^{-/-}$* cell lines stably carrying specific SAS-6 expression constructs were isolated in the absence of doxycycline (see Methods). Upon induction by doxycycline treatments (*Figure 1F*; *Figure 1—figure supplement 1D*), the ability of each SAS-6 mutant in driving de novo centrosome formation was examined (*Figure 1G*). Note that similar results were also seen in reconstitution experiments done in non-isogenic condition (*Figure 1E*; *Figure 1—figure supplement 2*). All SAS-6 fragments lacking the C-terminal domain (DM1, DM3, and DM5) failed to induce any centrosome formation (*Figure 1E,G*), although expressed at similar or higher levels (*Figure 1F*; *Figure 1—figure supplement 1D*). The C-terminal tail alone (DM6) was also insufficient. However, when the full-length coiled-coil domain (DM2) or a small portion of it (DM4) was present together with the C-terminal tail, it effectively drove de novo centrosome formation in all cells (*Figure 1E,G*; 100). Consistently, SAS-6 harboring the F131E mutation (F131E) that has been previously shown to disrupt the oligo-merization property of SAS-6 (*Kitagawa et al., 2011*; *van Breugel et al., 2011*) also efficiently drove de novo centrosome formation (*Figure 1E–G,*100%), with an expression level slightly higher than the endogenous level (*Figure 1F*; *Figure 1—figure supplement 1D*). A recent report showed that the fly SAS6 mutant equivalent to human SAS-6$^{F131E}$ could support the formation of some centriole-like structures in vivo (*Cottee et al., 2015*). Our result thus suggests that in contrary to the SAS-6 self-assembly model, the main activity of SAS-6 in promoting centriole assembly is possessed by its C-terminal portion, rather than the N-terminal domain mediating SAS-6 self-oligomerization.

## Centrioles formed in the absence of SAS-6 self-oligomerization can duplicate and ciliate

To determine whether centrioles form the core of these de novo centrosomes, SAS-6$^{-/-}$ cells arrested in S phase were induced to express various types of SAS-6 mutants, and examined for de novo centriole assembly (see 'Materials and methods'). Consistently, FL, F131E, DM2, and DM4, all of which contain intact C-termini, efficiently induced the formation of de novo centrioles in all cells (100%), as revealed by the co-localization of centrin, SAS-6, STIL, CEP135, and CPAP (*Figure 2*; *Figure 2—figure supplement 1*). Moreover, some of these de novo centrioles, even those formed with DM4 that lacks the entire N-terminal half of SAS-6, could later mature properly by acquiring distal appendages. Importantly, upon G1 arrest, some of these mature centrioles were able to support ciliogenesis (*Figure 2B*), suggesting that functionally normal centrioles can arise from de novo assembly through SAS-6 mutants that are incapable of self-oligomerization.

We next examine whether de novo centrioles formed with F131E or DM4 can later support canonical duplication in S phase after they have been converted to centrosomes (*Wang et al., 2011*). Strikingly, both F131E- and DM4-derived centrioles could duplicate, as revealed by centrin and STIL staining, with an efficiency grossly similar to or only slightly less than that of centrioles derived from wild-type SAS-6 (*Figure 2C,D*). This result, however, appears inconsistent with previous reports, where neither F131E nor DM4 could fully support centriole duplication (*Fong et al., 2014*; *Kitagawa et al., 2011*; *van Breugel et al., 2011*), although both were able to initiate SAS-6/cartwheel assembly in the presence of pre-existing centrioles (*Fong et al., 2014*). We do not understand the reason behind this discrepancy, but the result that F131E or DM4 can support the duplication of F131E- or DM4-derived centrioles, respectively (*Figure 2C,D*), but not pre-existing, wild-type SAS-6-derived centrioles (*Fong et al., 2014*; *Kitagawa et al., 2011*; *van Breugel et al., 2011*) raises an idea that perhaps pre-existing centrioles have an active, dominant role in guiding canonical duplication.

## Ninefold symmetric centrioles can form in the absence of SAS-6 self-oligomerization

As SAS-6 self-oligomerization is also thought to define the ninefold symmetric shape of centrioles, it is possible that de novo centrioles derived from F131E or DM4 can function normally as centrosomes or basal bodies, but their shape or structural integrity may be defective. To determine the structural integrity of de novo centrioles, serial sectioning transmission electron microscopy (TEM) was used to analyze SAS-6$^{-/-}$ cells inducibly expressing FL, F131E or DM4. To our surprise, normal looking centrioles, which are characterized as 200 nm x 500 nm in size, made of nine microtubule triplets, and equipped with distal and sub-distal appendages, could be easily found in cells expressing either F131E or DM4 (*Figure 3A–C*). In particular, EM tomography reconstruction of a centriole from DM4 expressing cells revealed a perfect ninefold symmetric shape (*Figure 3B*; *Video 1*). Moreover, EM analyses also confirmed that at least some DM4-derived centrioles could ciliate (*Figure 3C*) and duplicate (*Figure 3D*), firmly demonstrating that neither structural assembly nor shape determination of centrioles strictly depends on the self-oligomerization activity of SAS-6. Rather, the C-terminal half of SAS-6 is both required and sufficient for centriole biogenesis.

### De novo centriole assembly is an error-prone process

While SAS-6 self-assembly is not essential for centriole biogenesis, structurally normal centrioles can indeed form in the absence of pre-existing centrioles, questioning the necessity of having mother centrioles in canonical duplication. We thus asked if the defect-free rate of de novo centriole production is comparable to that of canonical duplication. EM analyses revealed that canonically duplicated centrioles are essentially free of error in their size, structural integrity, and geometric shape (*Figure 4E*, 100%, n = 70). In striking contrast, we consistently observed a small but significant portion of de novo centrioles that are abnormal, including centrioles missing variable numbers of MT triplets, or varying in size or shape (*Figure 4A–D*; *Video 2*). Importantly, similar types of abnormalities were seen regardless of whether centrioles were derived from SAS-6$^{FL}$, SAS-6$^{F131E}$, or SAS-6$^{DM4}$. Nevertheless, these abnormally looking centrioles appeared to have normal activities, suggesting that they are not simply unstable structures in the process of non-specific disintegration. For example, we observed SAS-6$^{FL}$-derived centrioles that had only six MT triplets, but able to duplicate,

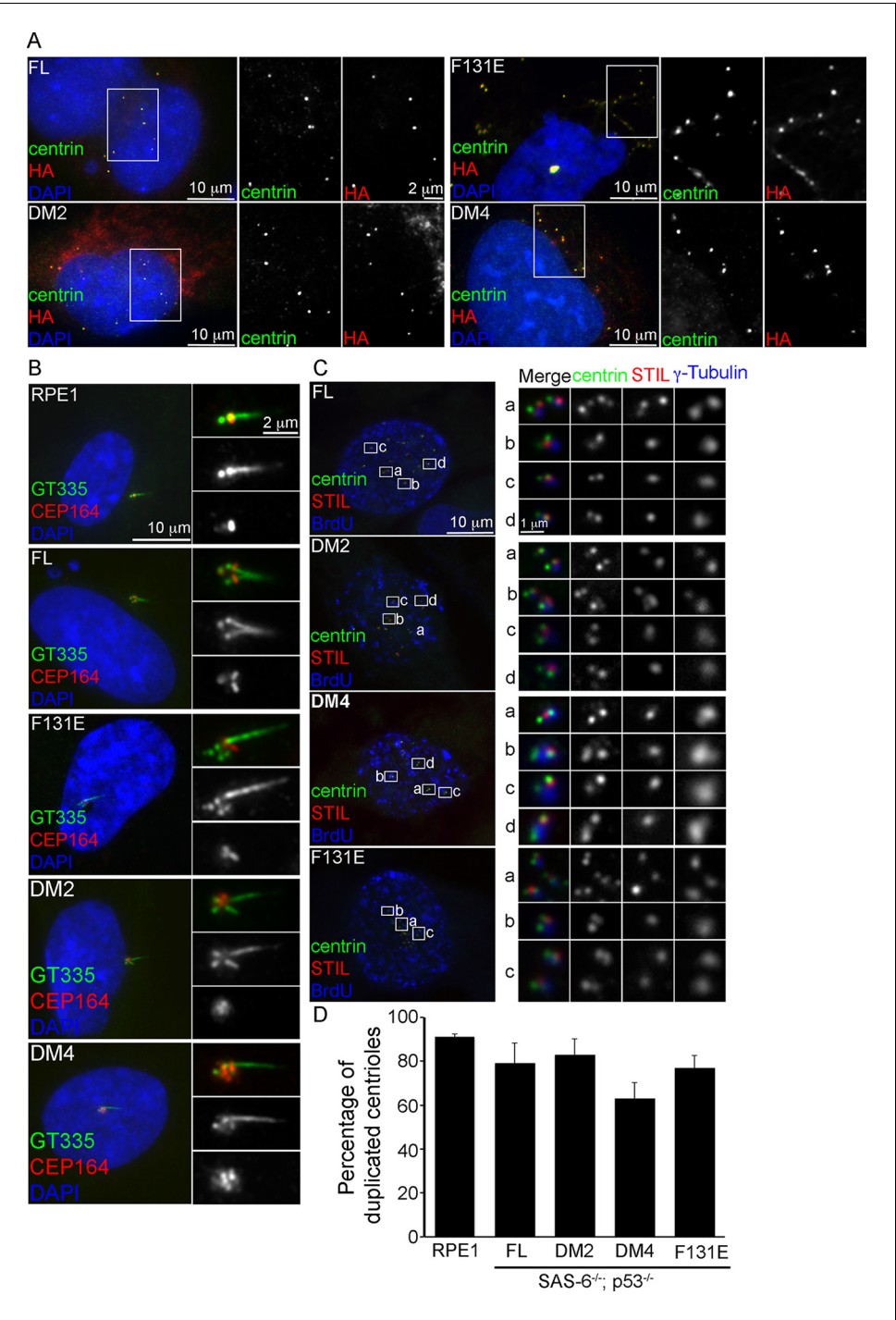

**Figure 2.** De novo centrioles formed in the absence of SAS-6 self-oligomerization can ciliate and duplicate. (**A**) Isogenic *SAS-6⁻/⁻; p53⁻/⁻* cells carrying indicated SAS-6 constructs (see 'Methods') were arrested in S phase and induced to express indicated SAS-6 mutants for 16 hr and then immunostained with indicated antibodies to visualize de novo centrioles (centrin, green; SAS-6/HA, red). DNA (DAPI, blue). (**B**) Isogenic *SAS-6⁻/⁻; p53⁻/⁻* cells carrying indicated SAS-6 constructs (see 'Methods') were induced to express indicated SAS-6 mutants for 3 days, arrested in G1 for additional 36 hr, and then processed for immunofluorescence microscopy to visualize cilia (GT335, green) and distal appendage (CEP164, red). DNA (DAPI, blue). (**C**) Isogenic *SAS-6⁻/⁻; p53⁻/⁻* cells carrying indicated SAS-6 constructs (see 'Methods') were induced to express indicated SAS-6 mutants for 3 days, and then processed for BrdU pulse-chase before fixation for immunofluorescence. Centriole duplication was revealed with indicated antibodies (centrin, green; daughter centriole marker, STIL, red; PCM marker, γ-tubulin, blue). S-phase cells labeled with BrdU were shown (blue). (**D**) Quantification of the centriole duplication efficiency from (**C**). S-phase cells were identified (BrdU+) and their centrioles were analyzed by immunofluorescence with centrin, STIL, and γ-tubulin antibodies. Error bars represent standard error of the mean (SEM); n > 150, N = 3.

*Figure 2. continued on next page*

*Figure 2. Continued*

The following figure supplements are available for Figure 2:

**Figure supplement 1.** Characterization of de novo centrioles formed in the absence of SAS-6 self-oligomerization.

producing daughter centrioles also made of an incomplete set of MT triplets (*Figure 4A-#1*). SAS-6$^{FL}$-derived centrioles that appeared larger than normal centrioles (*Figure 4A-#2*), but able to mature and acquire appendages were also found. In addition, we observed SAS-6$^{FL}$-derived centrioles made of disorganized MT triplets (*Figure 4A-#3*), or centrioles comprising of 9 MT triplets but existing as a distorted open cylinder (*Figure 4A-#6*). Consequently, the width (outer diameter) of SAS-6$^{FL}$-derived centrioles is more variable than that of canonically duplicated centrioles (*Figure 4D*). De novo centrioles made of disorganized, or abnormal numbers of, MT triplets have also been seen to form in cells depleted of the endogenous centrioles by laser ablation (*La Terra et al., 2005*).

Similarly, abnormal centrioles were observed in the population of F131E- or DM4- derived centrioles, including smaller/incomplete centrioles (*Figure 4B,C*), or bigger centrioles that were made of more than nine MT triplets (*Figure 4C-#1*). Notably, DM4-mediated, but not F131E-mediated, de novo synthesis clearly has an error rate higher than those mediated by SAS-6$^{FL}$ (*Figure 4E*), suggesting that the N-terminal half of SAS-6 is involved in some additional processes that ensure the accuracy or quality control of centriole assembly. Taken together, these results suggest that in the absence of pre-existing centrioles, de novo centriole synthesis can occur, largely independent of SAS-6 self-oligomerization, but the process is inherently error-prone, and may require the presence of pre-existing centrioles to achieve high accuracy.

## Discussion

We have established a cell-based system for reconstitution of de novo centriole assembly in human cells. Using this system, we found that self-oligomerization of SAS-6, an activity proposed to drive cartwheel/centriole formation, is in fact not essential for the structural assembly or shape determination of centrioles. Instead, de novo centriole formation can be sufficiently driven by the C-terminal half of SAS-6 lacking the self-oligomerization activity, fundamentally deviating from the current paradigm for centriole biogenesis. Moreover, our results show that in human cells, de novo centriole assembly is an error-prone process that generates abnormal centrioles in a significantly higher frequency comparing to canonical duplication. The error rate of de novo assembly, which was estimated from later stages of centrioles, could potentially be underestimated, as it is plausible that the de novo centrioles formed initially are actually much more error-prone, and that over time ninefold symmetric centrioles persist because they are more stable or more efficient to duplicate. In any case, our results suggest that if de novo assembly must be used for centriole production, the structural accuracy may need to be insured or reinforced by other mechanisms.

The expression level of the SAS-6$^{FL}$ or SAS-6$^{F131E}$ in our cell-based assay is higher than the endogenous level of SAS-6 responsible for canonical duplication (*Figure 1F*; *Figure 1—figure supplement 1D*), a condition that may potentially affect the accuracy of de novo centriole assembly. However, as de novo centriole assembly is normally blocked under physiological conditions (for number control), it is unclear what the 'right' level of SAS-6 should be, or if the level really matters that much, when the number control is not a relevant issue for de novo assembly. Consistently, it has been reported that de novo centrioles made of disorganized, or abnormal numbers of MT blades are formed with the endogenous level of SAS-6 (*La Terra et al., 2005*). Conversely, during multiciliogenesis where SAS-6 level is highly up-regulated (*Hoh et al., 2012*), almost every new centriole forms correctly. These results suggest that when cells carry pre-existing centrioles, they could perhaps tolerate a wider range of SAS-6 level for proper structural assembly of centrioles. Numerical control of centriole biogenesis, however, is a different issue, as it is highly sensitive to the sas-6 level (*Strnad et al., 2007*), even in the presence of pre-existing centrioles.

Our results thus lead to an interesting question: Is de novo synthesis a reliable, physiologically relevant pathway for centriole production in animals? Large quantities of centrioles naturally arising

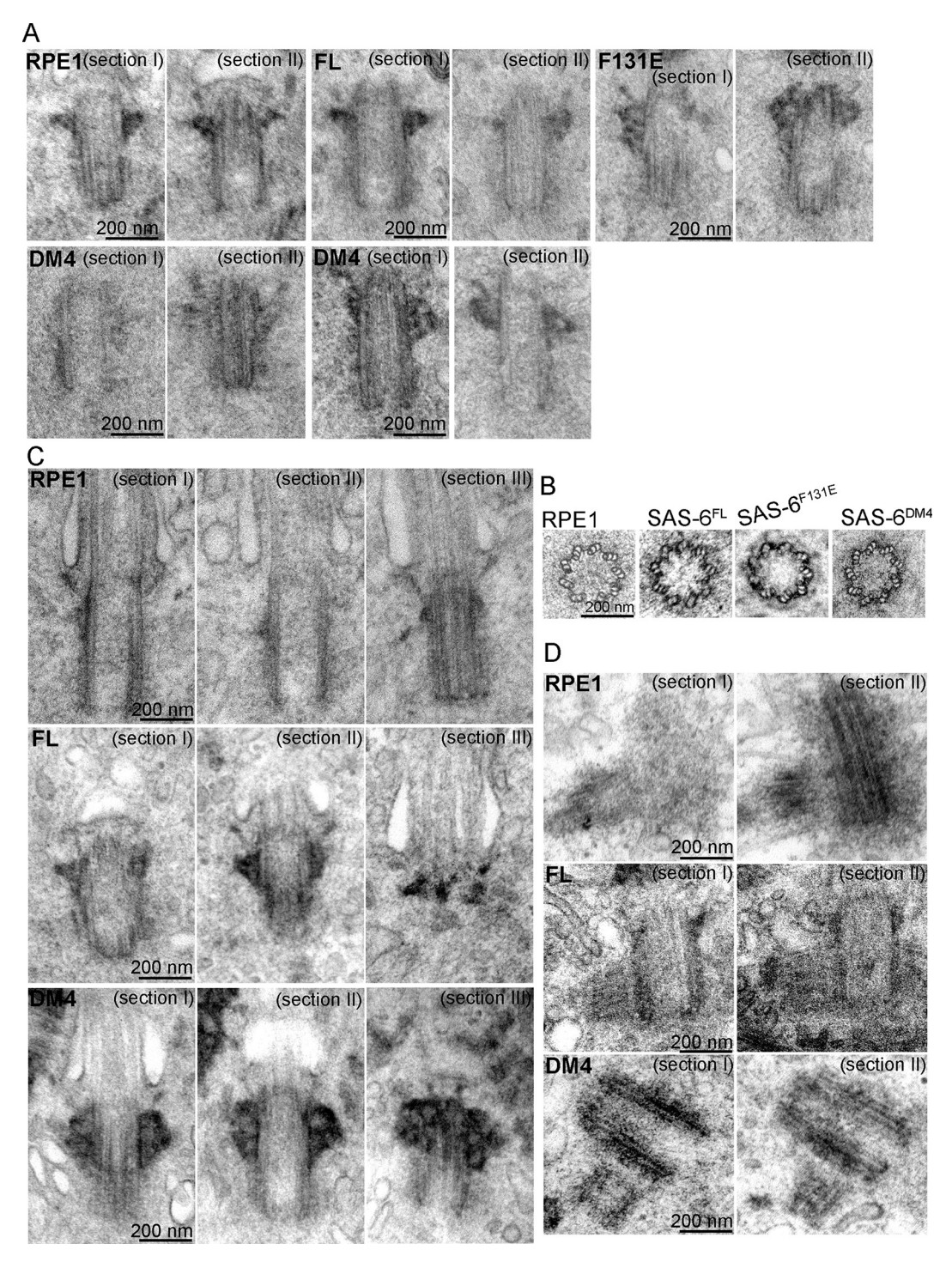

**Figure 3.** SAS-6 self-assembly is not essential for the ninefold symmetry of centrioles. Isogenic *SAS-6⁻/⁻; p53⁻/⁻* cells carrying indicated SAS-6 constructs (see 'Methods') were induced to express indicated SAS-6 mutants for 3 days, and then processed for serial sectioning electron microscopy. (**A,B**) Mature canonical centrioles in WT RPE1 cells, or de novo centrioles formed in SAS6⁻/⁻ cells expressing full-length (FL) or mutant SAS-6 as indicated are shown in longitudinal (**A**) or cross sectional (**B**) views. Note that these centrioles were mature and have acquired appendages. (**C**) Ciliated centrioles from indicated cell types were serially sectioned and examined by EM. (**D**) Duplicated, engaged centriole pairs from indicated cell types were serially sectioned and examined by EM. Scale bar: 200 nm.

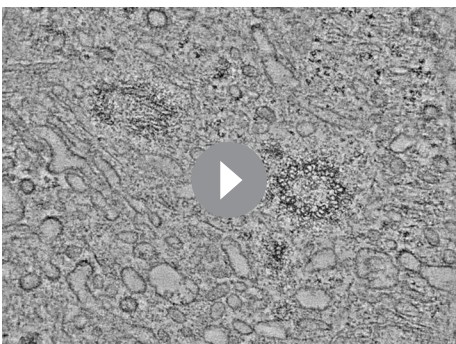

**Video 1.** EM tomography of a normal centriole from SAS-6[DM4] expressing cells (related to *Figure 3*). Note that a centriole on the right in cross-sectional views contains nine MT triplets.

from the acentriolar pathway has been reported in planarians (*Azimzadeh et al., 2012*), but whether the same process occurs in vertebrates under normal physiological conditions is unclear. It will be very interesting to see if planarians carefully specify the structural integrity of de novo centrioles, and if so, how the accuracy is achieved in the absence of pre-existing centrioles. In vertebrate somatic cycling cells, the birth of a new centriole is normally given by the pre-existing centriole (canonical duplication), a process known to regulate the number at which newborn centrioles can form. It has led to a belief that de novo assembly would take over the canonical duplication, when large, variable numbers of centrioles are needed, such as in the post-mitotic, multi-ciliated epithelium. Strikingly, a recent report showed that in multi-ciliated epithelia, de novo assembly is not responsible for the production of hundreds of centrioles (*Al Jord et al., 2014*); instead, all new centrioles that later form cilia are initiated, directly or indirectly, from the pre-existing centriole, supporting the idea that the presence of pre-existing centrioles has additional roles other than the number control for centriole reproduction. Our results that canonically duplicated centrioles are predominantly free of error, while de novo centriole assembly is error-prone, suggest that perhaps pre-existing centrioles can specify some steps of the assembly process that define the structural integrity of the centriole (e.g. cartwheel assembly), an idea consistent with the template-based model of centriole formation proposed recently (*Fong et al., 2014*).

Notably, mouse embryos show exceptions that break all the known rules. The mouse embryogenesis starts with a fertilized egg lacking apparent centrioles/centrosomes, and finishes with an intact animal in which nearly all cells carry centrioles that are both numerically and structurally normal (*Abumuslimov et al., 1994*; *Gueth-Hallonet et al., 1993*; *Howe and FitzHarris, 2013*; *Palacios et al., 1993*). How acentrosomal cells in mouse embryos escape from p53 dependent elimination acting against acentrosomal mitosis (*Bazzi and Anderson, 2014*; *Lambrus et al., 2015*; *Wong et al., 2015*), and how they later form and maintain proper centrioles in the absence of pre-existing centrioles are completely not understood. With our surprising observation that SAS-6 self-oligomerization is not essential for centriole assembly, perhaps other unanticipated mechanisms are specifically involved in centriole biogenesis in early mouse embryos.

## Material and methods

### Cell culture, SAS-6 constructs, and antibodies

Human telomerase-immortalized retinal pigment epithelial cells (hTERT-RPE1 or RPE1) were cultured in DME/F-12 (1:1) medium supplemented with 10% FBS and 1% penicillin-streptomycin. The expression constructs of the various SAS-6 DMs under the control of tetracycline-inducible promoter were made using the lentiviral vector pLVX-Tight-Puro vector (Clonetech) as described previously (*Fong et al., 2014*). Antibodies used in this study include anti-α-tubulin (Sigma-Aldrich), anti-HA.11 (Covance), anti-polyglutamylated tubulin (GT-335; Adipogen), anti-pericentrin and CPAP (Proteintech), anti-centrin2 (Millipore), anti-BrdU (AbD Serotec), CEP164 (Novus Biologicals), anti-STIL (Bethyl laboratories, Inc), anti–p53 and hSAS-6 (Santa Cruz Biotechnology; sc-6243, sc-376836 and

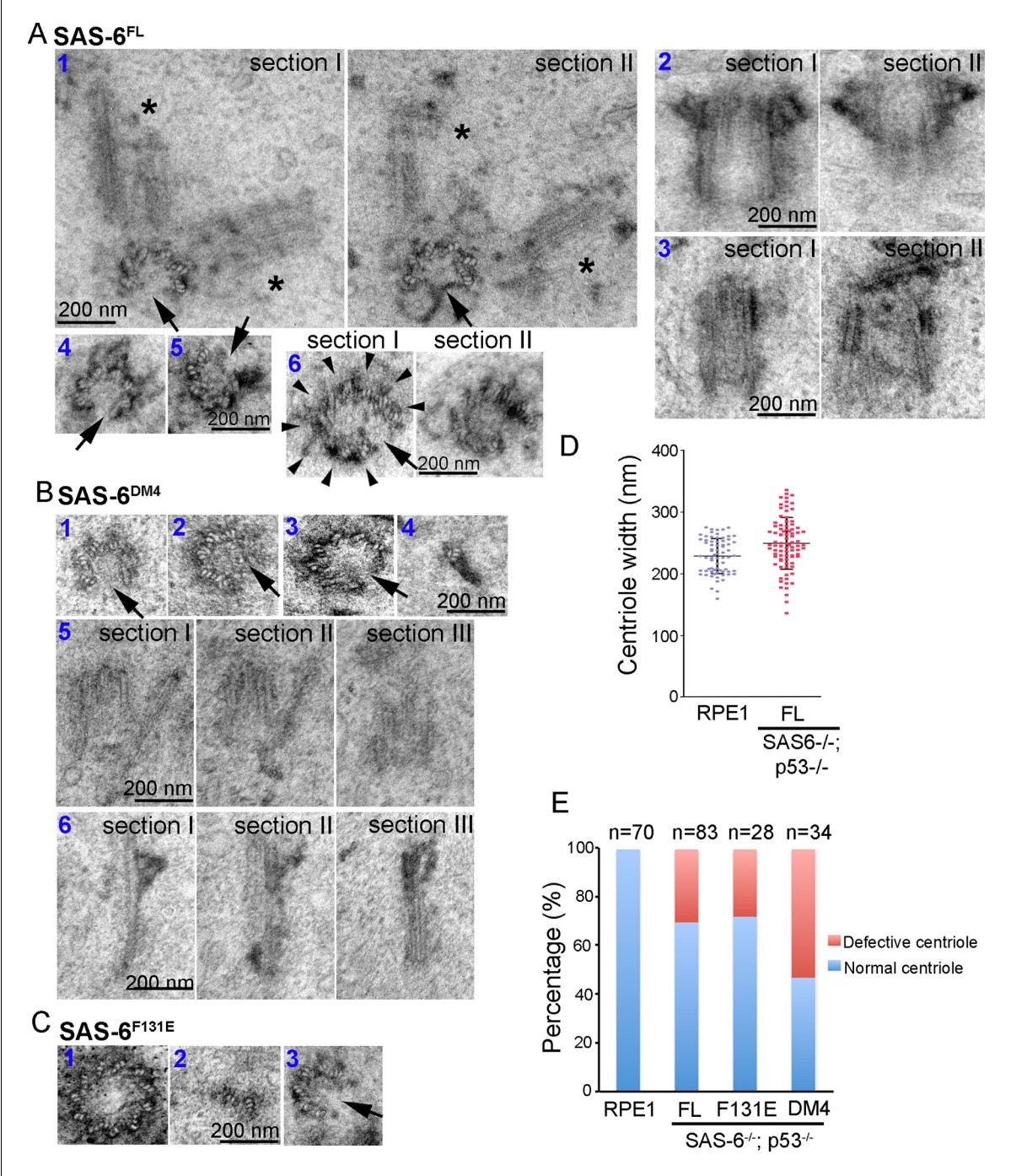

**Figure 4.** Centrioles formed through de novo assembly are error-prone. (**A–C**) Isogenic *SAS-6⁻/⁻; p53⁻/⁻* cells carrying indicated SAS-6 constructs (see 'Methods') were induced to express indicated SAS-6 mutants for 3 days, and then processed for serial sectioning electron microscopy. (**A1**) A mature, SAS-6^FL-derived centriole missed three MT triplets but was able to duplicate, producing daughter centrioles also made of an incomplete set of MT triplets (see stars in sections I and II). It appears that two daughter centrioles were formed at the same time. (**A2**) A mature, SAS-6^FL-derived centriole 33% wider than normal centrioles was able to acquire appendages. (**A3**) A SAS-6^FL-derived centriole made of disorganized MTs. (**A4&5**) Cross-sectional views of SAS-6^FL-derived, abnormal centrioles missing MT triplets. (**A6**) A SAS-6^FL-derived centriole carrying 9 MT triplets (arrow heads) but existing as a distorted open cylinder (arrow). (**B**) Abnormal centrioles derived from SAS-6^DM4 were shown in cross-sectional or longitudinal views. (**C**) Abnormal centrioles derived from SAS-6^F131E were shown in cross-sectional views. Note that a larger centriole made of 11 MT triplets was shown (**C1**). (**D**) The outer diameter of centrioles was quantified for both canonical centrioles (in normal RPE1 cells), and SAS-6^FL-derived de novo centrioles. (**E**) Quantifications showing the error rates of de novo and canonical centrioles. Sample size (n) is indicated. Note that arrows in all images indicate the missing of MT triplets.

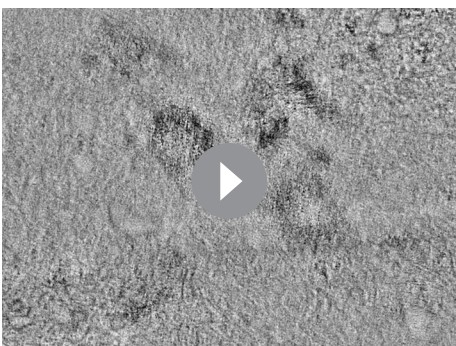

**Video 2.** EM tomography of an abnormal centriole from SAS-6^DM4 expressing cells (related to *Figure 4*). Note that a centriole on the right in cross-sectional views contains eight MT triplets.

sc-81431), and anti-CEP135 (abcam, ab75005). The antibody for N-terminal SAS-6 (sc-376836) is a monoclonal antibody raised against amino acids 1–300, with the actual epitope being mapped within amino acids 173–300 (*Figure 1—figure supplement 1D*). The antibody for C-terminal SAS-6 (sc-81431) is a monoclonal antibody raised against amino acids 404–657, with the epitope being mapped within amino acids 519–657 (*Figure 1F*).

## CRISPR construction and generation of *SAS6^-/-; p53^-/-* acentriolar cells

RNA-guided targeting of genes in human cells was achieved through coexpression of the Cas9 protein with gRNAs using reagents prepared by the Church group (*Mali et al., 2013*), which are available from the Addgene (http://www.addgene.org/crispr/church/). The targeting sequence for TP53 and SAS-6 is 5'-GGCAGCTACGGTTTCCGTC-3' and 5'-GTGAAATGCAAAGACTGTG-3', respectively, which were cloned into the gRNA cloning vector (Addgene plasmid #41824) via the Gibson assembly method (New England Biolabs, Ipswich, MA) as described previously (*Mali et al., 2013*). To obtain stable acentriolar cells lacking SAS-6, the TP53 gene in RPE1 cells was targeted by the CRISPR method prior to inactivation of SAS-6. Six days after SAS-6 inactivation, we observed that about 10–15% of cells were devoid of centrioles or centrosomes. Pure acentriolar cell lines were subsequently established through clonal propagation from single cells, a process taking additional 4–5 weeks (before these cells were used for experiments), generating a number of independent *SAS-6^-/-; p53^-/-* cell lines all behaving similarly (with clone #1 and #2 being characterized here). *SAS-6^-/-; p53^-/-* cells actively proliferate or divide, but take longer periods of time to go through mitosis (*Figure 1D*). For genotyping, the following PCR primers were used: 5'-ATCGGAATTCGGCCAAGTC-TCTTACGCCTT-3' and 5'- CTAGTCTAGAATGTGAGCCGGCTTCCTAAC-3' for SASS6 alleles, and 5'- ACGCGGATCCACCCATCTACAGTCCCCCTTG-3' and 5'-CTAGTCTAGAGCATCCCCAGGAGA-GATGCT-3' for TP53 alleles. PCR products were cloned and sequenced.

## Reconstitution of de novo centriole/centrosome formation

To examine the role of SAS-6 in de novo centriole formation, *SAS-6^-/-; p53^-/-* cell lines generated above were infected with lentiviruses carrying various of SAS-6 constructs, and induced to express wild-type or mutant SAS-6 with 50 ng/ml Doxycycline for 16 hr. To examine the function of de novo centrioles to form centrosomes, to duplicate, or ciliate, infected cells were incubated with doxycycline for 3 days, followed by serum starvation if ciliogenesis was to be examined. Isogenic, acentriolar cell lines stably carrying specific SAS-6 expression constructs (SAS-6-expression-ready cells) were isolated and propagated from single cells in the absence of doxycycline, which allow us to directly induce de novo centriole/centrosome formation with doxycycline addition. Our reconstitution of de novo centriole/centrosome formation was successfully done in acentriolar cells infected with viruses and then treated with doxycycline (*Figure 1E*; *Figure 1—figure supplement 2C,D*), or in isogenic, SAS-6-expression-ready cells treated with doxycycline (*Figures 1G,2*).

## Immunofluorescence and time-lapse microscopy

Cells were fixed with methanol at $-20°C$ for 5 min. Slides were blocked with 3% bovine serum albumin (w/v) with 0.1% Triton X-100 in PBS before incubating with the indicated primary antibodies. Secondary antibodies were from molecular probes and were diluted 1:500. DNA was visualized using 4′,6-diamidino-2-phenylindole (DAPI; Molecular Probes). Fluorescent images were acquired on an upright microscope (Axio imager; Carl Zeiss) equipped with $100\times$ oil objectives, NA of 1.46, and a camera (ORCA ER; Hamamatsu Photonics). For time-lapse experiments, hTERT-RPE1 cell (wild-type, P53$^{-/-}$ or p53$^{-/-}$; SAS-6$^{-/-}$) were imaged using a Zeiss Axiovert microscope configures with a 10X objective, motorized temperature-controlled stage, environmental chamber, and $CO_2$ enrichment system (Zeiss, Germany). Images were acquired and processed by axiovision software (Zeiss, Germany).

## Transmission electron microscopy

Cells grown on coverslips made of Aclar film (Electron Microscopy Sciences) were fixed in 4% paraformaldehyde and 2.5% glutaraldehyde with 0.1% tannic acid in 0.1 M sodium cacodylate buffer at room temperature for 30 min, postfixed in 1% $OsO_4$ in sodium cacodylate buffer for 30 min on ice, dehydrated in graded series of ethanol, infiltrated with EPON812 resin (Electron Microcopy Sciences), and then embedded in the resin. Serial sections (~90-nm thickness) were cut on a microtome (Ultracut UC6; Leica) and stained with 1% uranyl acetate as well as 1% lead citrate. Samples were examined on JEOL transmission electron microscope. Tomography data was collected using a JEOL 1400 Plus TEM operated at 120 kV. Double tilt series were recorded from –60° to 60° with 1° increments using SerialEM software (*Mastronarde, 2005*). Tomograms were reconstructed using IMOD programs (*Kremer et al., 1996*).

## Acknowledgements

We thank N Lampen at MSKCC for assisting the usage of transmission electron microscopy. This work is supported by the National Institutes of Health grants GM088253 and the American Cancer Society RSG-14-153-01-CCG to M-F B Tsou. WJ Wang is supported by the Taiwan Ministry of Science and Technology (MOST 103-2320-B-010-046-MY2).

## Additional information

### Funding

| Funder | Grant reference number | Author |
| --- | --- | --- |
| National Institutes of Health | GM088253 | Meng-Fu Bryan Tsou |
| American Cancer Society | RSG-14-153-01 | Meng-Fu Bryan Tsou |
| Ministry of Science and Technology, Taiwan | 103-2320-B-010-046-MY2 | Won-Jing Wang |

The funders had no role in study design, data collection and interpretation, or the decision to submit the work for publication.

### Author contributions

WJW, Conception and design, Acquisition of data, Analysis and interpretation of data, Drafting or revising the article, Contributed unpublished essential data or reagents; DA, Acquired EM tomography, Acquisition of data; CHK, WNJ, KU, Acquisition of data; MFBT, Conception and design, Analysis and interpretation of data, Drafting or revising the article, Contributed unpublished essential data or reagents

### Author ORCIDs

Won-Jing Wang, http://orcid.org/0000-0001-9733-0839

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
