## [Decision Letter]

Thank you for submitting your work entitled "De novo centriole formation in human cells is error-prone and does not require SAS-6 self-assembly" for peer review at *eLife*. Your submission has been favorably evaluated by Randy Schekman (Senior editor), a Reviewing editor, and three reviewers.

The reviewers have discussed the reviews with one another and the Reviewing editor has drafted this decision to help you prepare a revised submission.

Summary:

Centrioles are cylindrical, microtubule-based organelles that form the core of the centrosome and are essential for the formation of cilia and flagella. In cycling cells, centrioles duplicate exactly once in each cell cycle, through the formation of a single new daughter centriole on the wall of each existing mother centriole. The first morphological event in the assembly process is the emergence of a cartwheel structure adjacent to the wall of the mother centriole. SAS-6 proteins are a key structural component of the cartwheel. SAS-6 forms a parallel homodimer that associates laterally through a globular N-terminal domain. Models have shown that the association of the SAS-6 N-terminal domain can generate a ring of nine homodimers and this has been proposed to explain the ninefold symmetrical architecture of the cartwheel.

In this manuscript, Wang et al. test the widely-held assumption that SAS6 self-oligomerization is a necessary step in centriole assembly. The authors produce a p53/SAS6 knockout cell line that lacks centrioles, and subsequently introduce wild type or mutant SAS6 constructs to test which domains of SAS6 are required for de novo centriole assembly. Surprisingly, the authors find that mutants of SAS6 that are unable to undergo self-oligomerization are sufficient for assembling functional centrioles. Interestingly, centrioles that form de novo even in the presence of wild type SAS6 are frequently found to have structural defects. Most surprisingly, de novo centrioles assembled with SAS6 fragments that are unable to self-associate were capable of undergoing canonical duplication. Since the same SAS6 fragments are not capable of supporting the duplication of existing centrioles, the authors suggest that existing centrioles may play a role in shaping the structural integrity of procentrioles.

Essential revisions:

1) The interpretation of the work depends on the nature of the SAS6 deletion allele in the RPE1 cells used. If, for example, a fragment of protein is made, this would critically alter the interpretation. Thus, the authors should provide additional characterization of the p53/SAS6 knockout RPE1 cells that form the basis of this paper. Specifically, the DNA sequence of the edited SAS6 alleles should be provided, and possible predicted truncated products indicated. Furthermore, the authors should provide the product number for their SAS6 antibody, and describe the antigen that was used to make it. Ideally, they would test for the presence of SAS6 RNA, and test several SAS6 polyclonal antibodies on their knockout cell line to assure that a SAS6 fragment is not being produced from the putative knockout allele. The history of mammalian genetics is replete with examples of alleles thought to be null that were not indeed null – this is a critical point.

2) At the outset the authors don't quantify how the p53^-/-^; SAS6^-/-^ cells lose centrioles or show that they completely lack centrioles. Importantly, it is also unclear how long the cells used in these experiments have been cultured in the absence of centrioles and p53 before the expression of the Sas-6 constructs is induced. The authors state that clonal cell lines were derived for all constructs, but it is not clear how many were tested and how variable the expression levels were between different clonal lines. These points should be addressed in the text.

3) The size and number of de novo centrioles induced by SAS6 construct expression is not quantified or statistically analyzed (the authors show a single image of a few cells for each construct). This is important, as this seems to be quite variable and the authors make several statements implying that all of these de novo centrioles can duplicate and form cilia, which seems surprising given this variation. The authors should explain their criteria for scoring something as a centriole/centrosome/cilia, and show graphs of the number and size distribution of these organelles for all the expression constructs tested. This is symptomatic of a general problem of making statements that are too strong for the data presented. For example, the authors consistently state that all the de novo structures they observe are normal centrioles that can form normal centrosomes and cilia, but this seems very unlikely given the variation in the size and number of these structures formed in each cell (see Figure 1). In the text, for example, the authors state that "all these de novo centrioles" could mature properly by acquiring distal appendages, and support ciliogenesis upon G1 arrest. Are the authors really claiming that all of these structures form normal cilia? The cilia shown in Figure 2 look quite variable in size and number. As noted above, it would require proper quantitative serial section EM to prove these points, and the authors already know from their EM analysis that some of these structures are not normal, even without such quantitation. Importantly, even if these structures look normal at the EM level, it does not mean they are fully functional. The authors are urged to be more cautious here and to perhaps only claim that morphologically normal centriole-like structures can be formed in the presence of these constructs, rather than implying that all these de novo structures will form completely normal centrioles, centrosomes and cilia.

4) As the authors note, it is surprising that the majority of de novo centrioles formed in the presence of the SAS-6 constructs that cannot self-assemble through the head group have a normal structure, and this is the major point of the paper. The lack of quantitation in some of the EM experiments makes this difficult to assess. This is particularly important because it seems plausible that during the early stages of de novo assembly the process is actually very error-prone and that over time 9-fold symmetric centrioles persist because they are more stable (and perhaps cells with 9-fold symmetric centrioles are more viable). The authors should at least consider this possibility, especially in light of the Lambrus et al. (2015) paper, which would support this idea.

5) The authors should quantify by how much these SAS6 constructs are overexpressed, as the overexpression of SAS6 has previously been shown to drive the de novo formation of abnormal centriole-like-structures in fly eggs (Rodrigues-Martins et al., 2007 and Peel et al., 2007). From the single blot shown, it seems that the proteins are expressed at close to normal levels, but this should be quantified, and some estimate of the experimental variability should be given. This should also be assessed for all the experimental conditions tested (e.g. the authors don't assess SAS6 protein levels in the experiments where the cells are extensively arrested in S-phase or G1). This potential caveat to the interpretation of these results should be discussed.

6) The authors claim the de novo centrioles formed with any of the Sas-6 constructs can all undergo canonical duplication. Others have shown that when centrioles are re-introduced into cells lacking centrioles there is a burst of de novo assembly, but the cell population then stabilizes its centriole numbers and seems to revert to normal canonical duplication – e.g. Wong et al., 2015 (this paper should be referenced) and Lambrus et al., 2015. It would be very informative to know whether this happens for the constructs tested here. If the WT constructs can support the return to canonical duplication, while the F131E and DM4 constructs cannot, this would suggest that the mutant constructs can support some level of error-prone de novo assembly, but cannot support canonical duplication. Although it is not essential that the authors perform this experiment for the revision, if they have the data available, they would strengthen the paper.

7) Related to this last point, it is surprising that the authors don't discuss in any detail why their results might differ from those reported by other groups. One possibility is that they are assessing error-prone de novo assembly from a cell line that started with no centrioles, while the other groups were starting from cell lines that had centrioles and where they depleted the endogenous SAS6 in the presence of RNAi resistant forms of SAS6 that could not multimerize. Moreover, they should mention the recent observation that a form of fly SAS6 carrying the equivalent to the F131E mutation could also support the formation of some centriole-like structures in vivo (although these were not examined at the EM level – Cottee et al., 2015).

8) The authors frame their paper as a contrast between the self-assembly model and the templating model, but the two models are not mutually exclusive. As the authors note, it is surprising that mutants that are self-assembly deficient are remarkably efficient in de novo centriole formation, but this does not necessarily prove the self-assembly model wrong. Purified proteins may be self-assembly deficient in vitro, but be assisted by other proteins (restoring some oligomerization) in vivo. The templating model is attractive, but it does not readily explain the importance of the C-terminal end domains revealed in the de novo assembly system used here, where there is no mother centriole lumen with which to interact. The authors should adopt a more conciliatory tone with regard to the two previously proposed models. Once again, these are not mutually exclusive, and the present data, although very interesting and worthy of publication, do not allow us to identify a "winner."

---

## [Author Response]

We would like to thank the reviewers and editors for the positive notes and critical suggestions. We have revised the manuscript accordingly, and provided a separate file with tracked changes. Significant changes are itemized below, followed by our point-by-point responses that address reviewers’ concerns.

New items:

1) Figure 1—figure supplement 1: detailed characterization of the *sas-6^-/-^; p53^-/-^* cell lines (clone #1 and #2), including genotyping, determination of the (exogenous) SAS-6 expression level, and mapping of the SAS-6 antibodies (N- or C-terminal) we used.

2) Figure 1—figure supplement 2: characterization of de novo centriole/centrosome formation in clone #2 *SAS-6^-/-^; p53^-/-^* cell line.

3) Figure 2—figure supplement 1: the original Supplemental Figure 1.

4) Figure 4: during the revision, we found that the Figure 4-#6 and 4C-#3 in the original figure were mistakenly swapped. These images, which look quite similar, were misplaced during imaging preparation.

5) Figure 4: quantification of the width (outer diameter) of canonical and de novo centrioles.

6) Materials and methods: new details were added to subsection “CRISPR construction and generation of *SAS6^-/-^; p53^-/-^* ancentriolar cells”. In addition, a new subsection titled “Reconstitution of de novo centriole formation” is added to the Materials and methods, describing how we induce de novo centriole formation in clonal or non-clonal cell populations.

7. Other changes are detailed below in our point-by-point response.

*Essential revisions:*

*1) The interpretation of the work depends on the nature of the SAS6 deletion allele in the RPE1 cells used. If, for example, a fragment of protein is made, this would critically alter the interpretation. Thus, the authors should provide additional characterization of the p53/SAS6 knockout RPE1 cells that form the basis of this paper. Specifically, the DNA sequence of the edited SAS6 alleles should be provided, and possible predicted truncated products indicated. Furthermore, the authors should provide the product number for their SAS6 antibody, and describe the antigen that was used to make it. Ideally, they would test for the presence of SAS6 RNA, and test several SAS6 polyclonal antibodies on their knockout cell line to assure that a SAS6 fragment is not being produced from the putative knockout allele. The history of mammalian genetics is replete with examples of alleles thought to be null that were not indeed null – this is a critical point.*

We have now provided suggested characterizations on two independent *sas-6^-/-^; p53^-/-^* cell lines (clone #1 and #2). Both clones behave similarly for de novo centriole or centrosome formation (Figure 1 and Figure 1—figure supplement 2); detailed EM analyses of de novo centrioles were done only with cell lines derived from clone #1.

DNA sequence analyses showed that each allele of the SAS-6 gene in either clone #1 or #2 was edited to carry frameshift mutations in the beginning of the coding region (Figure 1—figure supplement 1), predicted to produce severely truncated products consisting of only 19 or 23 amino acids in clone #1 or #2 respectively. Consistently, no SAS-6 signal can be detected by western blot using antibodies against either the N- or C-terminal region of SAS-6 (Figure 1; Figure 1—figure supplement 1). The antibody for N-terminal SAS-6, from Santa Cruz (sc-376836), is a monoclonal antibody raised against amino acids 1-300, with the actual epitope being mapped within amino acids 173-300 (Figure 1—figure supplement 1; Materials and methods). The antibody for C-terminal SAS-6, from Santa Cruz (sc-81431), is a monoclonal antibody raised against amino acids 404-657, with the epitope being mapped within amino acids 519-657 (Figure 1; Materials and methods). We have also tried several other polyclonal antibodies raised against the N-terminus of SAS-6, including Novus (NB100-93342), Bethyl Labs (A301-801A), and Santa Cruz (sc-98506), but none of them worked in our hands. In addition to SAS-6, we also found and mapped frameshift mutations in TP53 alleles, leading to loss of p53 function (Figure 1—figure supplement 1).

*2) At the outset the authors don't quantify how the* p53^-/-^; SAS6^-/-^
*cells lose centrioles or show that they completely lack centrioles. Importantly, it is also unclear how long the cells used in these experiments have been cultured in the absence of centrioles and p53 before the expression of the Sas-6 constructs is induced. The authors state that clonal cell lines were derived for all constructs, but it is not clear how many were tested and how variable the expression levels were between different clonal lines. These points should be addressed in the text.*

a) More information regarding how we isolate acentriolar cell lines has been added (in the main text and Materials and methods). In brief, six days after SAS-6 inactivation by CRISPR treatments (in p53^-/-^ background), we observed that 10-15% of cells in the population were devoid of centrioles or centrosomes. Pure acentriolar cell lines were further isolated through clonal propagation from single cells, a process taking additional four to five weeks, before these cells were used for reconstitution experiments (see b) below). That is, our acentriolar cell lines have been devoid of SAS-6 proteins or centrioles for at least 6 weeks or longer before they were used for reconstitution experiments.

b) Regarding whether our observation occurs only in one particular line (due to clonal variation), we have now made it clear in the text and Materials and methods that de novo centriole assembly could be efficiently driven by FL, F131E, DM2, and DM4 in either mixed or isogeneic cell populations. That is, when the two independent SAS-6^-/-^ cell lines (clone #1 and #2) were infected with lentivirus carrying DOX-inducible SAS-6 constructs (FL, F131E, DM2, or DM4) and incubated with DOX, we observed de novo centriole/centrosome formation in every infected cells expressing detectable SAS-6 (Figure 1; Figure 1—figure supplement 2 C, D). Based on these results, we then isolated stable acentriolar cell lines carrying different SAS-6 constructs in the absence of DOX (SAS-6-expression-ready cells), and used these isogenic lines for subsequent reconstitution experiments for a better control on timing and protein expression. Importantly, again, the ability of FL, F131E, DM2, or DM4 to drive de novo centriole formation is consistently seen in both isogenic and non-isogenic conditions.

c) The SAS-6 expression level in each isogeneic cell line we used has now been quantified and shown (Figure 1 and Figure 1—figure supplement 1), which is somewhat higher than the endogenous level of SAS-6 that is responsible for canonical duplication. Note that the “proper” level of SAS-6 for de novo centriole assembly is unknown, as in human body, no known de novo assembly pathway exists under physiological conditions (our discovery here may explain why it shouldn’t exist). Please see the response 5a) below for more discussion.

*3) The size and number of de novo centrioles induced by SAS6 construct expression is not quantified or statistically analyzed (the authors show a single image of a few cells for each construct). This is important, as this seems to be quite variable and the authors make several statements implying that all of these de novo centrioles can duplicate and form cilia, which seems surprising given this variation. The authors should explain their criteria for scoring something as a centriole/centrosome/cilia, and show graphs of the number and size distribution of these organelles for all the expression constructs tested. This is symptomatic of a general problem of making statements that are too strong for the data presented. For example, the authors consistently state that all the de novo structures they observe are normal centrioles that can form normal centrosomes and cilia, but this seems very unlikely given the variation in the size and number of these structures formed in each cell (see Figure 1). In the text, for example, the authors state that "all these de novo centrioles" could mature properly by acquiring distal appendages, and support ciliogenesis upon G1 arrest. Are the authors really claiming that all of these structures form normal cilia? The cilia shown in Figure 2 look quite variable in size and number. As noted above, it would require proper quantitative serial section EM to prove these points, and the authors already know from their EM analysis that some of these structures are not normal, even without such quantitation. Importantly, even if these structures look normal at the EM level, it does not mean they are fully functional. The authors are urged to be more cautious here and to perhaps only claim that morphologically normal centriole-like structures can be formed in the presence of these constructs, rather than implying that all these de novo structures will form completely normal centrioles, centrosomes and cilia.*

We apologize for our ambiguous writing. We absolutely did not mean to say that every de novo centriole formed in our assay could mature and support ciliogenesis. Our intention was to indicate that *all* SAS-6 constructs that drive de novo centriole assembly, including F131E & DM4, can produce centrioles capable of supporting ciliogenesis (or duplication and centrosome formation). This is supported by both IF (with centriole/PCM/cilia markers), and EM. We have fixed our writing to make it very clear that only some (not all) of the de novo centrioles can duplicate and ciliate.

The number of de novo centriole formed in each cell is variable within the population for all relevant SAS-6 constructs. We think this is the nature of de novo assembly, and decided not to focus on it, as the number can be affected by many (known or unknown) factors that could not be tightly controlled in our assay. Our main point in this study is that every SAS-6^-/-^ cell can produce some numbers of de novo centrioles when certain SAS-6 mutants are re-introduced.

*4) As the authors note, it is surprising that the majority of de novo centrioles formed in the presence of the SAS-6 constructs that cannot self-assemble through the head group have a normal structure, and this is the major point of the paper. The lack of quantitation in some of the EM experiments makes this difficult to assess. This is particularly important because it seems plausible that during the early stages of de novo assembly the process is actually very error-prone and that over time 9-fold symmetric centrioles persist because they are more stable (and perhaps cells with 9-fold symmetric centrioles are more viable). The authors should at least consider this possibility, especially in light of the Lambrus et al. (2015) paper, which would support this idea.*

Thanks for the suggestion; we have included it in the Discussion.

*5) The authors should quantify by how much these SAS6 constructs are overexpressed, as the overexpression of SAS6 has previously been shown to drive the de novo formation of abnormal centriole-like-structures in fly eggs (Rodrigues-Martins et al., 2007 and Peel et al., 2007). From the single blot shown, it seems that the proteins are expressed at close to normal levels, but this should be quantified, and some estimate of the experimental variability should be given. This should also be assessed for all the experimental conditions tested (e.g. the authors don't assess SAS6 protein levels in the experiments where the cells are extensively arrested in S-phase or G1). This potential caveat to the interpretation of these results should be discussed.*

a) We have provided SAS-6 quantification, shown in Figure 1 and Figure 1—figure supplement 1. The western blot was done with SAS-6^-/-^ cells inducibly expressing various SAS-6 mutants for 3 days, during which cells were allowed to proliferate. This is the condition and time point where we analyzed centrioles with EM. The expression levels of FL and F131E are similar to each other, but both slightly higher than the endogenous SAS-6 seen in normal RPE1 cells. The potential caveat of SAS-6 over-expression has been included in the Discussion. However, as de novo centriole assembly is normally blocked under physiological conditions (for number control), we really do not known what the “right” level of SAS-6 should be for this pathway, and not sure whether the level matters that much, especially when the number control is not a relevant issue for de novo assembly. Consistently, it has been reported that de novo centrioles made of abnormal numbers of MT blades are formed with endogenous levels of SAS-6 (La Terra et al., 2005). Moreover, during multiciliogenesis where the SAS-6 level is highly up-regulated, almost every new centriole still forms correctly, suggesting that when cells carry preexisting centrioles, they could perhaps tolerate a wider range of SAS-6 level for proper structural assembly of centrioles. Numerical control of centriole biogenesis, however, is a different story, as it is highly sensitive to the sas-6 level (Strnad et al., 2007), even in the presence of preexisting centrioles. This is also included in the Discussion.

b) The structures formed in unfertilized fly eggs upon overexpression of SAS-6 indeed carry some centriole-like features at the IF/LM level (e.g. able to recruit some centriolar or PCM markers), but at the EM level, they are not centrioles. Unlike cycling cells, unfertilized eggs naturally do not support centriole assembly, and are unique in many other ways; we are reluctant to compare these two systems as it is not the subject of this study, but we clearly show that normal looking centrioles capable of duplication and ciliation can form in our reconstitution assay.

*6) The authors claim the de novo centrioles formed with any of the Sas-6 constructs can all undergo canonical duplication. Others have shown that when centrioles are re-introduced into cells lacking centrioles there is a burst of de novo assembly, but the cell population then stabilizes its centriole numbers and seems to revert to normal canonical duplication – e.g. Wong et al., 2015 (this paper should be referenced) and Lambrus et al., 2015. It would be very informative to know whether this happens for the constructs tested here. If the WT constructs can support the return to canonical duplication, while the F131E and DM4 constructs cannot, this would suggest that the mutant constructs can support some level of error-prone de novo assembly, but cannot support canonical duplication. Although it is not essential that the authors perform this experiment for the revision, if they have the data available, they would strengthen the paper.*

a) We apologize for missing the key reference, and have now included it.

b) We examined centriole duplication 3 days after induction SAS-6 expression, and saw that most cells had a high number of centrioles/centrosomes due to the initial burst of de novo assembly, and more importantly, most of these centrioles could duplicate (based on STIL & centrin doublet staining) regardless of whether they were derived from FL, F131E or DM4. Similar results were also seen by EM. We do notice that centriole numbers drop significantly after a long-term culture, but we are not sure whether this is due to an efficiency issue (i.e. less centrioles are easier to maintain), or alternatively, due to a fitness issue (i.e. a selection against the survival of cells carrying higher numbers of centrioles). This question is very interesting but complicated, requires additional careful investigations, and thus should be answered in a different paper. Our goal is to address whether FL, F131E or DM4 derived centrioles can duplicate, and the answer is clearly yes for at least a significant fraction of centrioles at day 3 (based on IF and EM).

*7) Related to this last point, it is surprising that the authors don't discuss in any detail why their results might differ from those reported by other groups. One possibility is that they are assessing error-prone de novo assembly from a cell line that started with no centrioles, while the other groups were starting from cell lines that had centrioles and where they depleted the endogenous SAS6 in the presence of RNAi resistant forms of SAS6 that could not multimerize. Moreover, they should mention the recent observation that a form of fly SAS6 carrying the equivalent to the F131E mutation could also support the formation of some centriole-like structures in vivo (although these were not examined at the EM level – Cottee et al., 2015).*

a) Similar to the reviewers’ idea, we did propose that preexisting centrioles have some active, dominant roles in guiding canonical duplication (at the end of Figure 2 result). We think this idea could perhaps explain why F131E or DM4 can support the duplication of F131E- or DM4-derived centrioles, respectively (Figure 2), but not wild-type SAS-6-derived centrioles (Fong et al., 2014; Kitagawa et al., 2011; van Breugel et al., 2011). That is, only F131E- or DM4-derived centrioles but not WT centrioles can use SAS-6^F131E^ or SAS-6^DM4^, respectively, to duplicate.

b) The paper by Cottee et al. has been added.

*8) The authors frame their paper as a contrast between the self-assembly model and the templating model, but the two models are not mutually exclusive. As the authors note, it is surprising that mutants that are self-assembly deficient are remarkably efficient in de novo centriole formation, but this does not necessarily prove the self-assembly model wrong. Purified proteins may be self-assembly deficient in vitro, but be assisted by other proteins (restoring some oligomerization) in vivo. The templating model is attractive, but it does not readily explain the importance of the C-terminal end domains revealed in the de novo assembly system used here, where there is no mother centriole lumen with which to interact. The authors should adopt a more conciliatory tone with regard to the two previously proposed models. Once again, these are not mutually exclusive, and the present data, although very interesting and worthy of publication, do not allow us to identify a "winner."*

The two main points of this study are (i) SAS-6 self-oligomerization (but not self-assembly of other proteins, or self-assembly in general) is not essential for centriole formation, and (ii) de novo but not canonical centriole assembly is error prone. Both points are clearly stated in the title, Abstract, and Results; no other points beyond these are concluded. While we are not sure which statements the reviewers were referring to, we have removed any irrelevant words that may potentially cause misunderstanding.

In the Discussion, however, we feel that it is necessary to discuss how the template-based model can fit in with the new results we have discovered here.